# Stall-Feeding of Sheep on Restricted Grazing: Effects on Performance and Serum Metabolites, Ruminal Fermentation, and Fecal Microbiota

**DOI:** 10.3390/ani13162644

**Published:** 2023-08-16

**Authors:** Mengyu Zhao, Xiaoan Zhang, Yao Chen, Chunhuan Ren, Yiming Sun, Penghui Wang, Xiao Cheng, Zijun Zhang, Jiahong Chen, Yafeng Huang

**Affiliations:** 1College of Animal Science and Technology, Anhui Agricultural University, Hefei 230036, China; z13103756297@163.com (M.Z.); zxa1716424@163.com (X.Z.); c2062757209@163.com (Y.C.); renchunhuan@ahau.edu.cn (C.R.); sunyiming1150@163.com (Y.S.); wangpenghui15987@163.com (P.W.); chengxiao316@163.com (X.C.); zhangzijun@ahau.edu.cn (Z.Z.); chenjiahong@ahau.edu.cn (J.C.); 2Center of Agriculture Technology Cooperation and Promotion of Dingyuan County, Chuzhou 233200, China; 3Yingshang Agricultural Green Development Promotion Center, Fuyang 236200, China

**Keywords:** restricted grazing time, lambs, growth performance, serum metabolites, ruminal fermentation, rectal microbial

## Abstract

**Simple Summary:**

The global demand for mutton products has increased due to changes in consumer lifestyles, consumption concepts, and socio-economic developments. The feeding systems are widely acknowledged to significantly affect the growth, development, and body health of livestock. In this study, we evaluated the effects of different feeding methods on growth performance, serum biochemical indexes, rumen fermentation parameters, and fecal microorganisms of Huang-huai sheep. The results show that indoor feeding with restricted grazing artificial grassland can improve fattening performance, antioxidation and immune function activity, and rumen fermentation, as well as change fecal microbiota composition.

**Abstract:**

This study investigated the effects of three feeding systems, indoor feeding (CONT), indoor feeding with time-restricted grazing artificial pasture (4 h/day, G4H), and indoor feeding with an eight-hour daily grazing artificial pasture (G8H), on the growth performance, serum metabolites, ruminal fermentation, and fecal microbiota composition of lambs. Average daily gain showed a tendency (*p* = 0.081) to be higher for the G4H group compared with the CONT group. Moreover, feeding systems did not have a significant effect on most of the serum biochemical indicators in lambs. Concentrations of serum glutathione peroxidase and immunoglobulins (IgA, gG, and IgM) were significantly lower (*p* < 0.01) in the CONT group. Additionally, a tendency towards higher levels of volatile fatty acids, acetate, and butyrate was found in animals of the G4H group compared to the CONT group. Furthermore, fecal microbiota composition was altered in G4H and G8H groups, resulting in the increased relative abundance of *Firmicutes* and *Ruminococcaceae UCG-005*, as well as the decreased relative abundance of *Ruminobacter* compared with the CONT group. Overall, these results suggest that indoor feeding with restricted grazing time does not significantly affect fattening performance or rumen fermentation but enhances antioxidation and immune function activity and also alters fecal microbiota composition.

## 1. Introduction

The global demand for mutton products has increased due to changes in consumer lifestyles, consumption concepts, and socio-economic developments [1,2]. Moreover, mutton products are sought after for their low cholesterol and fat content, rich nutrition, unique flavor, and easy digestion and absorption [3,4]. The feeding systems are widely acknowledged to significantly affect the growth, development, and overall body health of livestock [5,6,7,8]. Grazing on pasture has been recognized as a cost-effective method for lamb production, providing a healthy and green meat that is highly valued by consumers [9]. However, it has been shown that this feeding system results in a 13.8% reduction in final body weight (FBW) and a 28.7% decrease in average daily gain (ADG) compared to indoor-fed lambs [10]. In contrast, lambs grazing on natural pastures and supplemented with 150–300 g of concentrate exhibited a 32.5% increase in ADG compared to indoor-fed counterparts [11]. Conversely, lambs grazing on pastures and supplemented with concentrate based on 0.8% of BW had a 27.3% decrease in FBW compared to stall-fed lambs [12]. Therefore, the above studies suggest that grazing pastures combined with dietary supplementation can improve lamb growth performance, and the level of supplementation also affects mutton fattening performance.

China is the world’s largest producer of mutton, with a population of 319.69 million sheep and goats and a meat production of 5.14 million tons in 2021. As policies restricting grazing on degraded grassland have been implemented, the traditional grazing system is being replaced by a semi-grazing and semi-housing system or even an intensive system [13,14,15]. However, due to the lack of physical exercise in indoor feeding systems and the prohibition of antibiotic addition to feed, livestock immunity has declined considerably, leading to reduced production performance and elevated mortality rates [16,17]. In this context, the strategy of indoor supplementation combined with restricted grazing time has been proposed as a more ecological strategy in lamb production, which has also been demonstrated to lead to optimal production performance and better health [11,18,19]. Previous studies had shown that lambs grazing for 4 h daily and receiving indoor supplementation exhibited significantly higher quality and productivity of the grassland compared to those grazing for 12 h per day only [11]. Chen et al. [20] also demonstrated that restricted grazing for four hours per day combined with supplementation with concentrate had a lesser negative impact on grassland vegetation, which is advantageous for the overall growth and sustainable utilization of grassland resources. Moreover, lambs subjected to restricted grazing (4 h) and indoor feeding exhibited similar FBW and ADG compared to those in the indoor feeding system, which leads to healthier meat with higher protein and n-3 polyunsaturated fatty acids (PUFA) contents and lower saturated FA n-3/n-6 PUFA ratio [11,21,22]. In addition, changes in feeding regimes can affect rumen fermentation characteristics and serum metabolites directly, with consequences for health and nutrient utilization, ultimately affecting fattening performance and meat quality [16,23,24]. Furthermore, the gastrointestinal microbiota of ruminants also plays a crucial role in animal physiology, nutrient digestion, body health, and productivity [25,26,27]. However, limited information is available regarding the effects of restricted grazing on artificial grassland combined with supplementary diets on the growth performance and health status of sheep as assessed by serum metabolite profiling, ruminal fermentation parameters, and gastrointestinal microbial population composition.

Based on this, the purpose of this study was to determine the effects of different feeding systems on growth performance, serum biochemistry, antioxidant and immunological profiles, and ruminal fermentation and fecal microbiota of Huang-huai sheep. To achieve this, three feeding systems were constituted: indoor feeding (CONT), indoor feeding with time-restricted (4 h) access daily to grazing artificial pasture (G4H), and (iii) indoor feeding with 8 h access to grazing artificial pasture (G8H). The findings discussed herein provide a theoretical basis and a practical reference for improving production performance and promoting sustainable and healthy development in the sheep industry.

## 2. Materials and Methods

The feeding trial was conducted from 3 April to 14 July 2021 at the Huangzhu Animal Husbandry Co., Ltd. based in Chuzhou city, China (32°57′ N, 117°50′ E, 71.7 m average above sea level). Animals were managed, fed, and reared in accordance with the guidelines of the Institutional Animal Care and Use Committee of Anhui Agricultural University (permit number SYXK 2016-007).

### 2.1. Animals and Experimental Design

Twenty-four female Huang-huai sheep, approximately 5 months old with an initial body weight of 28.96 ± 1.04 kg (mean ± SD), were randomly chosen and assigned to three treatment groups (eight lambs per group) as follows: (i) control group (CONT), in which animals received stall-feeding with concentrate and whole corn silage; (ii) animals had time-restricted access (4 h per day) to artificial mixed pasture complemented with concentrate and whole corn silage (G4H); and (iii) conventional eight-hour per day access to artificial mixed pasture complemented with concentrate and whole corn silage (G8H).

Lambs in G4H and G8H groups grazed artificial, mixed grassland, mainly composed of alfalfa (*Medicago sativa*), white clover (*Trifolium repens*), orchard grass (*Dactylis glomerata*), perennial ryegrass (*Lolium perenne* L.), and other pasture plants. The grassland was divided into four grazing plots (around 2800 m^2^/plot), which were grazed in a 15-day grazing rotation to minimize the confounding effects of sward characteristics. Animals in CONT and G4H groups were fed ad libitum total mixed ration (TMR) provided as two daily meals (8:30 a.m. and 6:30 p.m.); animals in G8H were fed ad libitum TMR provided as a single daily meal (6:30 p.m.); animals in G4H grazed from 9:00 a.m. to 1:00 p.m. during the adaptation period and 45 days prior to the fattening period, and grazed from 2:00 p.m. to 6:00 p.m. during the subsequent 45 days after the fattening period due to high environmental temperature at noon which affects feeding; animals in G8H grazed from 8:30 a.m. to 12:30 p.m. and from 2:00 p.m. to 6:00 p.m. The diets were composed based on feeding standards for meat-producing sheep and goats [28]. The experiment described in this study started in April and lasted for 108 days, with a twelve-day adaptation period, a ninety-day fattening period, and a six-day collecting trial for fecal samples. All animals had free access to fresh drinking water throughout the study. The ingredients and chemical composition of the prepared diets are shown in Table 1.

### 2.2. Growth Performance Traits

During the fattening period, lambs were weighed every 30 days prior to their morning feeding. Average daily gain (ADG, g/d) was calculated as weight gain divided by the number of days.

### 2.3. Serum Collection and Serum Analysis

After the end of fattening period, jugular vein blood samples (approximately 10 mL) were collected from each lamb before the morning feeding using vacuum tubes without anticoagulant. Then, the blood was clotted at room temperature for 30 min and centrifuged at 3000× *g* for 10 min at 4 °C to obtain serum which was stored at −80 °C until further analysis. Serum biochemical indices, including total protein (TP), albumin (ALB), globulin (GLB), urea nitrogen (UN), alanine aminotransferase (ALT), aspartate aminotransferase (AST), total cholesterol (TC), triglycerides (TG), high-density lipoprotein cholesterol (HDL-C), and low-density lipoprotein cholesterol (LDL-C) were determined using the biochemistry assay kits (Shenzhen Mindray Bio-Medical Electronics Co., Ltd., Shenzhen, China). Serum immunoglobulin G (Ig G), IgA, IgM, and glutathione peroxidase (GSH-Px) activity were measured using the enzyme-linked immunosorbent assay (ELISA) kits (Nanjing Jiancheng Bioengineering Institute, Nanjing, China), following the manufacturer’s instructions.

### 2.4. Ruminal Fermentation Analysis

At the end of the fattening period, approximately 30 mL of ruminal fluid samples per animal were collected from six lambs randomly selected from each treatment using an orogastric tube (transesophageal sampler: Flora Rumen Scoop, Wuhan, China) within 2–4 h after the morning feeding [29]. Ruminal pH was measured immediately after sampling using a portable pH meter (PHS-3C, Leici Scientific Instrument Co., Ltd., Shanghai, China). After filtering using four layers of cheesecloth, 10 mL of ruminal fluid sample was acidified by adding 2 mL of 25% HPO3 and stored at −20 °C for the determination of volatile fatty acids (VFA). Another 5.0-mL subsample of ruminal fluid sample was collected and stored at −20 °C for the determination of ammonia nitrogen (NH_3_-N) content. After thawing, ruminal fluid samples were centrifuged at 15,000 r/min for 15 min at 4 °C. Then, contents of ruminal VFA were determined by gas chromatography (GC-2010 Plus, Shimadzu, Japan) according to the method described by Huang et al. [30]. Ruminal NH_3_-N content was determined based on a colorimetric method [31] and adapted for Thermo Scientific Multiskan Go (BIO-RAD, Hercules, CA, USA) with a 700 nm absorbance filter.

### 2.5. Fecal Sampling, Microbiota Sequencing and Analysis

During the last six days of the experiment, fecal samples were collected. A three-day adaptation period was followed by three days of daily fecal sample collection occurring between 08:00 a.m. and 06:30 p.m. [32]. Fecal samples were manually collected from the rectum of six randomly selected lambs from each experimental group. The samples were immediately frozen in liquid nitrogen and stored at −80 °C until DNA extraction.

To determine fecal microbiota composition, high-throughput sequencing was performed at Majorbio Bio-Pharm Technology Co., Ltd. (Shanghai, China). Microbial DNA was extracted from fecal samples using the QIAamp DNA Stool Mini Kit (Qiagen, Shanghai, China) according to the manufacturer’s instructions. DNA concentration and quality were determined based on OD-1000+ (one drop, Shanghai, China) and 2% agarose gel electrophoresis. The V3-V4 regions of the bacterial 16S rRNA gene were amplified using primers 338F (5′-ACTCCT ACG GGA GGC AGC A-3′) and 806R (5′-GGA CTA CHV GGG TWT CTA AT-3′). PCR amplifications were conducted in a 20-μL mixture containing 4 μL of 5 × FastPfu buffer, 0.2 μM of each primer (5 μM), 0.4 μL of FastPfu polymerase, 2 μL of 2.5 mM dNTPs and 10 ng of template DNA. PCR thermocycling conditions were as follows: 95 ℃ for 2 min, followed by 28 cycles at 95 ℃ for 30 s, 55 ℃ for 30 s, and 72 ℃ for 30 s, and a final extension at 72 °C for 5 min. PCR products were submitted to electrophoresis on 2% (*w*/*v*) agarose gels and recovered using an AxyPrep DNA Gel Extraction Kit (Axygen, Hangzhou City, China). Amplicons were then sequenced on an Illumina Miseq platform (Illumina, San Diego, CA, USA; Illumina Novaseq6000) using paired-end sequencing (2 × 250 bp).

Raw sequencing data were processed and analyzed by Trimmomatic V0.39 software [33]. Operational taxonomic units (OTUs) were clustered using UPARSE software (version 7.1) based on 97% similarity, and chimeric sequences were identified and removed. Venn analysis was performed on the number of OTUs obtained. Representative sequences of OTUs were aligned to the SILVA database for bacteria taxonomic assignments using QIIME (http://qiime.org/scripts/assign_taxonomy.html, accessed on 27 September 2022) [34]. Observed OTUs and Shannon index were used to assess alpha diversity in each fecal sample. Mann–Whitney test and Wilcoxon signed-rank test were used to analyze the variance in microbiota data.

Richness estimates and diversity indices, including Chao 1, phylogenetic diversity whole tree (PD whole tree), Good’s coverage, observed features, Shannon’s index, and Simpson’s index, were calculated using the QIIME V1.8 pipeline. Principal coordinate analysis (PCoA) based on weighted UniFrac distances was conducted to compare all samples, and a distance-based matrix analysis was performed to evaluate differences among samples. A distance-based matrices analysis (PERMANOVA) was conducted using the vegan package in the QIIME V1.8 software [35]. The bar graph of Kyoto Encyclopedia of Genes and Genomes (KEGG) function was generated using GraphPad Prism 6.0 software.

### 2.6. Statistical Analysis

Statistical analyses of indices related to growth performance, serum parameters, ruminal fermentation, and fecal microbiota were analyzed using a general linear model routine in SPSS software (Version 25.0, SPSS Inc., Chicago, IL, USA). The effect of feeding system on all variables was analyzed with the statistical linear model: Y_i_ = μ + D_i_ + E, where Y_i_ is the dependent variable, μ is mean value, D_i_ means the fixed effect of feeding system, and E means the random effect [36]. Significance was determined at a level of *p* ≤ 0.05. The means of each trait were determined using Duncan’s multiple comparison test if a significant effect of treatment was declared (*p*_treatment_ ≤ 0.05). Spearman’s correlation was used to evaluate the correlations among bacterial populations, ADG, serum metabolites, and rumen VFA.

## 3. Results

### 3.1. Growth Performances

During the three months of the experiment, lambs gained 15 kg BW on average. For the BW, no significant differences were found throughout the first month (*p* = 0.555), the second month (*p* = 0.416), and the average level (*p* = 0.235). Regarding the ADG, a tendency of significance was found throughout the first month (*p* = 0.069) and the overall average level (*p* = 0.080), and there was a tendency to higher ADG in the G4H group compared to the CONT group. However, the ADG in the second and last month of the experiment did not differ (*p* > 0.05) among the treatment groups and averaged 131.91 and 125.08 g/day, respectively (Table 2).

### 3.2. Serum Biochemical Parameters

The data on serum biochemical indices are shown in Table 3. No significant differences (*p* > 0.05) were found for serum biochemical indices among experimental groups, including ALB, GLB, ALT, AST, TC, TG, and LDL-C, with respective means 61.22 g/L, 23.02 g/L, 38.20 g/L, 16.66 IU/L, 101.86 IU/L, 1.81 g/L, 0.27 g/L, and 0.80 g/L. However, the ALB/GLB ratio (*p* = 0.087) and UN content (*p* = 0.096) showed an upward tendency in the G8H group compared with the CONT group. Moreover, LDL-C content was higher (*p* < 0.05) in the CONT group than in G4H or G8H group, with no significant difference between G4H and G8H groups.

### 3.3. Serum Antioxidant and Immunity Indices

The average content of GSH-Px in the serum was higher (*p* < 0.001; 36.70%) in G4H and G8H groups compared with that in the CONT group (Table 4). Moreover, significant differences (*p* < 0.01) were found for IgA, IgG, and IgM levels among the feeding regimes. The average levels of IgA, IgG, and IgM followed a similar trend and were found at similar levels in the serum of both G8H and G4H groups, but which were higher than in the CONT group (*p* < 0.01).

### 3.4. Ruminal Fermentation Parameters

The ruminal pH was found within the range of 6.28–6.65 and did not differ significantly (*p* > 0.05) among treatments (Table 5). Ruminal NH_3_-N content averaged 13.98 mg/dL and did not differ among treatments (*p* > 0.05). No difference was found among treatments for total VFA content (*p* > 0.05), but an upward trend was found for total VFA content in the G4H group (*p* = 0.10). For individual VFAs, the contents of propionate, valerate, isobutyrate, isovalerate, and acetate/propionate did not differ among treatments. However, a trend towards higher contents of acetate (*p* = 0.057) and butyrate (*p* = 0.071) was observed for the G4H group, respectively.

### 3.5. Fecal Bacterial Microbiome

#### 3.5.1. Bacterial OTUs and Alpha Diversity

A total of 955,355 bacterial reads were detected after filtering out low-quality reads of 18 fecal samples from lambs in three treatment groups. Among them, 332,681 bacterial reads were obtained from the CONT group; 321,722 bacterial reads from the G4H group; and 300,952 bacterial reads from G8H, a total of 24,435 OTUs were identified based on a similarity of 97%. As shown in the Venn diagram (Figure 1), 1792 OTUs were common to the three treatment groups.

After analysis using QIIME Pipeline, alpha diversity indices showed that there was no significant difference in Chao 1, PD_whole_tree, Goods_coverage, Observed_features, Shannon, and Simpson among the three treatment groups (Table 6). PCoA based on unweighted UniFrac beta-diversity indices showed that the CONT group was significantly separated from the other two groups; G4H and G8H groups had a partial crossover, and the difference between groups was significant (ANOSIM = 0.26, *p* < 0.05, Figure 2).

#### 3.5.2. Relative Abundance of Bacteria in Lamb Feces at the Phylum and Genus Level

Figure 3 shows the composition of fecal bacterial communities and details of the intergroup variations in the top 15 bacterial phyla and genera in terms of abundance. At the phylum level, the relative abundances showed *Firmicutes*, *Bacteroidota*, *Proteobacteria*, *Spirochaetota,* and so forth. The relative abundance of *Firmicutes* in the CONT group was significantly (*p* < 0.05) lower than in G4H and G8H groups, while no significant difference was found between G4H and G8H groups (*p* > 0.05; Figure 3a,b; Appendix A). The relative abundance of *Lentisphaerae* showed only a tendency (*p* = 0.091) to be lower for the G4H group compared with the G8H group.

At the genus level, feeding systems significantly impacted three bacterial genera such as *Ruminococcaceae UCG-005*, *Ruminococcaceae UCG-010,* and *Ruminobacter* (*p* < 0.05; Figure 3c,d; Appendix A). The relative abundance of *Ruminococcaceae UCG-005* in the CONT group was significantly lower than that in G4H and G8H groups, while that of *Ruminobacter* in the CONT group was significantly higher than that in G4H or G8H group. However, no significant difference was found between G4H and G8H groups. The relative abundance of the *Christensenellaceae R-7* group was significantly higher in the G4H group than in the CONT group, with the G8H group being intermediate (*p* = 0.052). Interestingly, the relative abundance of *Ruminococcaceae UCG-010* in the G8H group was significantly higher than that in CONT and G4H groups, although no difference was found between CONT and G4H groups. The relative abundance of *Prevotellaceae UCG-001* showed only a tendency (*p* = 0.075) to be higher for the G4H group compared with the G8H group.

#### 3.5.3. Prediction of Bacterial Gene Functions

The functional prediction of the fecal microbiota in lambs submitted to different feeding regimens is shown in Figure 4a,b. The identified genes were assigned to amino acid metabolism, carbohydrate metabolism, vitamin metabolism, and other metabolic pathways. Thus, it can be assumed that the microbiota is involved in physiological activities such as signal transduction, immunity, digestion, and excretion.

A comparative analysis of KEGG function prediction (Figure 5) showed that all functional genes that have no significant difference (*p* > 0.05) among the three treatment groups are related to amino acid metabolism, carbohydrate metabolism, vitamin metabolism, and other metabolic pathways.

#### 3.5.4. Correlation Analysis

It was found that the bacteria communities were related to the ADG (0–90 d), serum metabolites, and rumen VFA. *Ruminococcaceae UCG-005* was positively correlated with GSH-Px, IgM, VFA, and acetate but negatively correlated with UN (Figure 6). *Ruminococcaceae UCG-010* was positively correlated with Ig G and acetate. *Ruminobacter* was negatively correlated with GSH-Px. *Christensenellaceae R-7 group* and *Prevotellaceae UCG-001* were positively correlated with ADG.

## 4. Discussion

### 4.1. Growth Performances

The ADG and final BW were important indicators of growth performance in livestock and had a direct impact on the income of sheep farmers. Previous studies had shown that 6-month-old weaned male Barbarine lambs grazing on native pastures supplemented with concentrates had similar ADG and FBW compared with indoor-fed lambs [40]. Another study showed that weaned Hulun buir lambs of similar weight (17.00 ± 1.55 kg) grazing on natural grassland supplemented with concentrate within a limited timeframe (4 h/day) had similar final BW and tended (*p* = 0.100) to have higher ADG than those under stall-feeding conditions [41]. Similar to the above results, our study found that compared with indoor-fed lambs, ADG and final BW were similar under grazing supplemented with diets, while time-restricted grazing and indoor supplemented diets exhibited comparable FBW and tended to exhibit higher ADG than indoor-fed lambs. This growth-promoting effect of time-restricted grazing and indoor supplemented diets might be attributed to the natural behavior of sheep, which was in accordance with animal welfare and also contributes to enhancing grazing efficiency and high-quality fresh pasture consumption within a limited time compared to indoor roughage-fed animals, which in turn was conducive to more pronounced growth. Rehemujiang et al. [18] also demonstrated that the energy consumption of ruminants during grazing increased with the duration of grazing. However, Wang et al. [11] reported that 3-month-old male Tan lambs under time-limited grazing on natural pasture and supplemented with concentrate showed similar ADG and FBW values compared with indoor-fed animals. These results suggested that improving grazing grassland types (artificially mixed grasses vs. natural pasture) and increasing the amount of daily supplementary feed (twice vs. once a day) may enhance fattening performance.

### 4.2. Serum Biochemical Parameters

Serum biochemical indices are crucial for assessing the metabolic status of nutrients metabolism and the health status of livestock [24,42]. In this study, all biochemical parameters, with the exception of ALB and ALT levels in three treatment groups, were found within the reference range [37,38,39]. These results might suggest a favorable health status for the sheep. A decreased ALB/GLB ratio indicates an improved immune status in animals [43], while a lower UN content indicates higher nitrogen utilization efficiency [44]. In the present study, the trend towards a higher ratio of ALB/GLB and UN content was observed for G4H and G8H groups. These could be attributed to the increased physical activity of grazing lambs, which in turn promoted carbohydrate and fat degradation but enhanced protein synthesis and metabolism [45]. Additionally, increased physical activity also stimulates the proliferation and differentiation of lymphocytes, which boosts the immune function of the body [46].

The ALT and AST were important transaminases in animal tissue cells, particularly in the liver and heart muscle [47]. This study found that different feeding systems did not affect the levels of ALT and AST, indicating that changes in feeding systems did not damage the liver of livestock. It had been reported that yak calf grazing on natural pure pastures had increased ALT and AST levels compared with stall-fed animals [48]. Such differences in serum ALT and AST levels across different studies might be attributed to differences in feeding practices, including pure grazing versus supplementation, as well as variations in breeds.

Lipid metabolism-related parameters reflected the body’s lipid metabolic function, which was closely associated with growth, development, and body health [49]. LDL-C was the main transport form of lipids, mostly delivered as cholesterol to the liver for cholic acid synthesis. In this current study, a decrease in serum LDL-C levels was observed in G4H and G8H groups, possibly due to their higher level of physical activity as well as the lipid-lowering effect of bioactive factors (mainly saponins and flavonoids) found in alfalfa from grazing pasture [49]. Moreover, the absence of differences in contents of TP, ALB, GLB, TC, TG, and HDL-C among the three treatment groups suggested that indoor-fed and adding grazing methods might not have been sufficient to influence these biochemical parameters.

### 4.3. Serum Antioxidant and Immunity Indices

Oxidative stress was considered a primary factor contributing to various diseases in animals [50,51]. GSH-PX was an important peroxidase that indirectly reflects the body’s ability to scavenge free radicals. An increase in GSH-PX content was beneficial in reducing damage caused by peroxides [47]. The results of this study showed that lambs in G4H and G8H groups enhanced serum GSH-Px content on day 90, indicating that indoor-fed and adding grazing methods enhanced serum antioxidant enzymatic activity and improved the antioxidant status of lambs. This could be attributed to increased levels of physical activity, the diverse floristic composition of pastures, and the presence of alfalfa. In fact, Su et al. [49] also reported that dietary supplementation with alfalfa powder increased the GSH-Px activity in sheep serum.

Serum immunoglobulins, including IgA, IgG, and IgM, play a vital role in the immune function of animals [52]. The contents of serum IgA, IgG, and IgM in lambs in the G4H and G8H groups were clearly higher than those of lamb in the CONT group. These effects could be associated with increased physical activity and alignment with the natural behavior and welfare of lambs, which enhanced their immune function and promoted antibody production. These findings were consistent with previous studies by Qi et al. [16], who reported higher contents of serum IgA, IgG, and IgM in grazing pigs compared to indoor-fed pigs. Hu et al. [53] also found that chickens raised outdoors had significantly higher contents of serum IgA and IgM compared to those raised indoors.

### 4.4. Ruminal Fermentation Parameters

Ruminal pH was a crucial parameter of the stability of the inner rumen environment in ruminants [54,55]. In this study, ruminal pH was similar among treatment groups, ranging from 6.28 to 6.65, which fell within the normal range (5.5–7.0) for rumen fermentation [56,57], indicating the inner rumen environment of all lambs was in a normal state of fermentation. However, other studies reported significantly higher ruminal pH in yak calves grazing on pasture compared to indoor-fed animals [58]. The variations observed among these studies might be attributed to differences in feeding practices, such as exclusively grazing on natural pastures versus grazing on pastures supplemented with corn silage diets, as well as variations in animal breeds.

The contents of VFAs and NH_3_-N were the main internal environmental parameters of rumen fermentation [59]. In the present study, lambs in the G4H group exhibited a tendency towards higher content of total ruminal VFA compared to those in the CONT group. This suggested that reducing grazing time and providing a supplementary diet resulted in an increased energy supply to support BW gain [60,61], which might be attributed to the ability of the rumen system to adapt to different feeding systems by adjusting rumen microorganisms. This finding was similar to a study by Xue et al. [62], which reported that Tibetan sheep or yaks grazing on natural rangeland improved total VFA content compared to stall-fed animals. In the present study, the content of butyrate tended (*p* = 0.071) to increase in the G4H group compared with the G8H group. These effects could be associated with the highly fermentable non-fiber carbohydrates in more intake of supplementary diets, which promote ruminal butyrate production [63]. Moreover, concentrations of valerate, isobutyrate, and isovalerate did not differ among the three treatment groups. This may be attributed to their relatively minor contribution to the total VFA pool, and any changes in acetate or propionate concentrations could obscure their effects.

### 4.5. Fecal Bacterial Microbiome

The present experiment showed that alpha-diversity indices of the fecal microbiota in lambs of G4H and G8H groups were similar compared to the CONT group. This indicated that time-restricted grazing with supplementation or free grazing combined with supplementation had no significant effects on diversity indices compared to indoor feeding systems. However, samples pertaining to the CONT group clustered significantly, separated from the other two sample groups, indicating that feeding systems could influence the composition of bacteria colonizing sheep intestinal tract. Moreover, the dominant bacteria in lamb fecal samples were *Firmicutes*, *Bacteroidota*, and *Proteobacteria*, which was consistent with findings obtained in Tibetan sheep [32] and dairy cows [64].

Bacteria within the phylum *Firmicutes* were mainly involved in fiber decomposition and metabolism. The experiment showed the relative abundance of *Firmicutes* was higher in the G4H or G8H group than in the CONT group. The variation was likely driven by the change in the quality of dietary fiber intake from different feeding systems. *Ruminococcaceae_UCG-005* and *Ruminococcaceae UCG-010* was a member of the *Ruminococcus* family, which might facilitate the degradation of insoluble plant cell walls in the intestinal tract [65]. In the present study, the relative abundance of *Ruminococcaceae_UCG-005* was higher in G4H or G8H group compared to the CONT group, which might indicate an improved ability to degrade cellulose and hemicellulose due to the combination with grazing, thus resulting in higher dietary fiber intake, as *Ruminococcus* had a substantial set of hemicellulase and oligosaccharide-degrading enzymes [59]. Moreover, *Ruminococcus* bacteria played an important role in short-chain fatty acid production [66] and immunity in ruminants [27]. The increased abundance of these microorganisms might be one reason for the increased trend in total VFA generation and enhanced immune capacity in G4H or G8H group. *Ruminobacter* was involved in biohydrogenation processes in the rumen [67]. The present experiment showed that the relative abundance of *Ruminobacter* in lamb feces was higher in the CONT group than in G4H or G8H group, suggesting time-restricted grazing with supplementation or free grazing combined with supplementation might have the potential to improve biohydrogenation processes in the intestine.

Furthermore, the predicted functional profiles of the microbiota in lamb feces showed no significant alterations in response to different feeding systems. This indicated that different feeding systems might change the abundance of bacteria in lamb fecal microbiota, although no influence was observed on the overall function of the microbiotas. In consistency with our results, Zhang et al. [32] reported that lambs grazing on pasture exhibited similar functional profiles of the lamb fecal microbiota compared to those submitted to indoor feeding systems, despite changes in diversity, richness, and abundance of the bacterial community in lamb feces.

The gut microbiota provided a crucial role in defending against invasion by pathogenic microorganisms and ADG [27]. In this study, we found several bacterial taxa had a strong correlation with the growth, immunocompetence, and rumen VFA of the host. The genera from the family *Ruminococcaceae (Ruminococcaceae UCG-005* and *Ruminococcaceae UCG-010*) were positively correlated with the serum levels of immune markers and rumen short-chain fatty acids in the lambs. These genera potentially play an important part in the establishment process of host immunity and in producing short-chain fatty acids for lambs [68]. In contrast, genera *Ruminobacter* was negatively correlated with the serum levels of GSH-Px in the lambs, while *Prevotellaceae UCG-001* and *Christensenellaceae R-7 group* correlated positively with ADG of lambs. Further studies on the microbial ecology and metabolism of these bacteria will reveal the mechanistic basis of their association with host health and growth performance, which can facilitate their potential application as therapeutic agents or additives to promote animal growth.

## 5. Conclusions

The present findings revealed that lambs grazing on pasture daily for 8 h combined with supplementation did not impact ADG and final BW of lambs but improved serum immunity and antioxidant indices compared to indoor-fed animals. However, lambs grazing on pasture for 4 h daily with supplementation did not show a significant decrease in fattening performance and rumen fermentation indices, as well as significantly enhanced antioxidation and immunity abilities compared to indoor-fed animals. Moreover, the different feeding systems did not affect the alpha-diversity indices of fecal microbial populations in lambs. However, animals grazing on pasture either for 4 h or 8 h exhibited an increased relative abundance of *Firmicutes* and *Ruminococcaceae UCG-005* and decreased relative abundance of *Ruminobacter*, compared with indoor-fed animals. Consequently, restricted grazing time and an increased level of supplemental diets tend to promote fattening performance, significantly enhance antioxidation and immune function activity, as well as change the composition of fecal microbiota. Further studies are necessary to determine the effects on other important parameters for lamb production, such as ruminal microbiota composition, ruminal microbial protein synthesis, carcass attributes, and meat quality.

## Figures and Tables

**Figure 1 animals-13-02644-f001:**
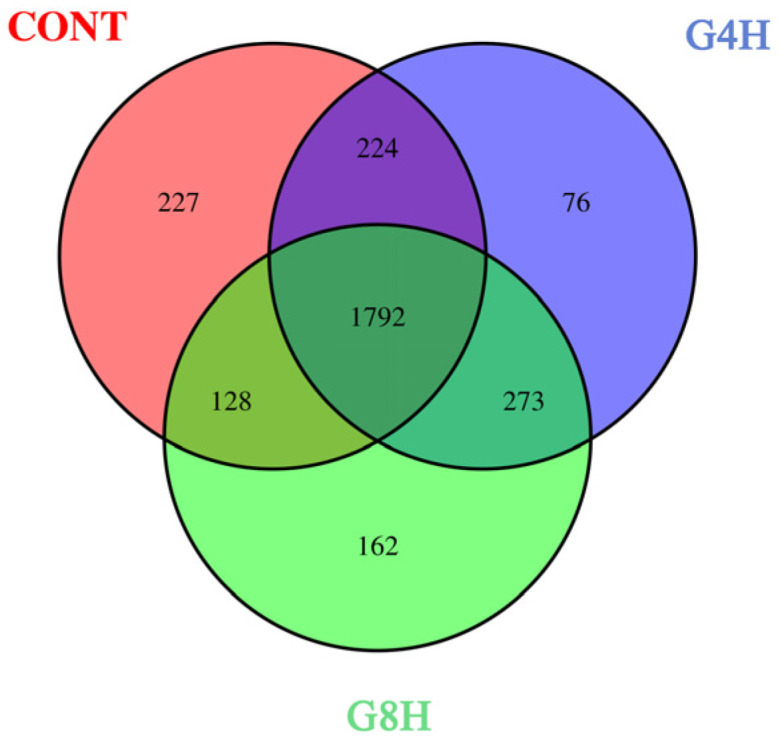
Venn analysis of operational taxonomic units (OTUs) of the fecal microbiota of sheep submitted to different feeding regimes. CONT, indoor-fed lambs without access to pasture; G4H, indoor-fed lambs with daily access to pasture for four hours; G8H, indoor-fed lambs with daily access to pasture for eight hours.

**Figure 2 animals-13-02644-f002:**
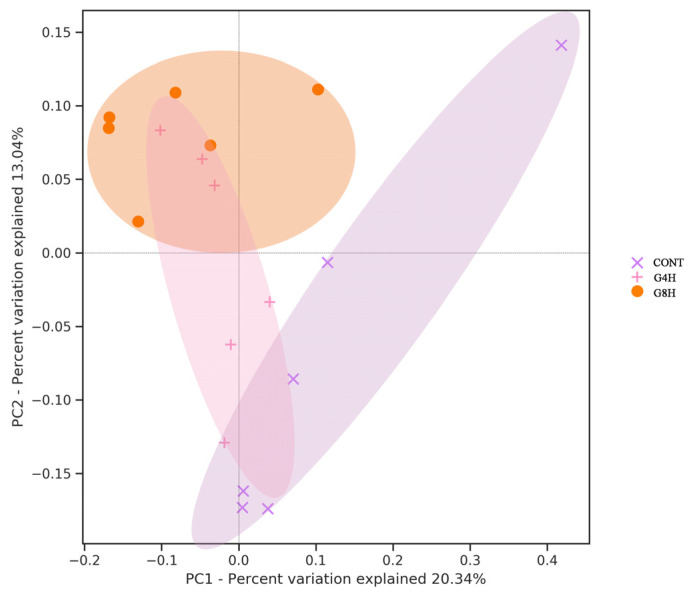
Principal coordinate analysis of the beta diversity of the fecal microbiota of sheep submitted to different feeding regimes. CONT, indoor-fed lambs without access to pasture; G4H, indoor-fed lambs with daily access to pasture for four hours; G8H, indoor-fed lambs with daily access to pasture for eight hours.

**Figure 3 animals-13-02644-f003:**
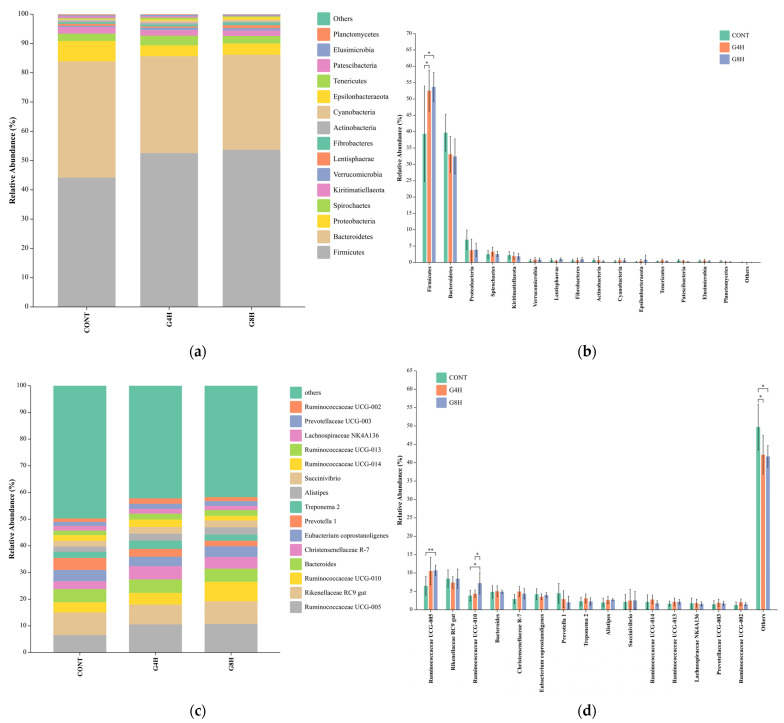
Composition of fecal bacteria at the phylum (**a**,**b**) and genus (**c**,**d**) of the rectal bacterial communities from different feeding regimes. CONT, indoor-fed lambs without access to pasture; G4H, indoor-fed lambs with daily access to pasture for four hours; G8H, indoor-fed lambs with daily access to pasture for eight hours, * 0.01 < *p* ≤ 0.05, ** 0.001 < *p* ≤ 0.01.

**Figure 4 animals-13-02644-f004:**
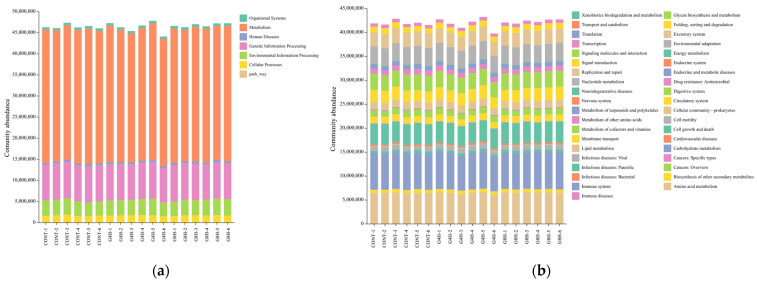
KEGG functional annotation by primary (**a**) and secondary (**b**) pathway of the rectal bacterial communities from different feeding regimes. CONT, indoor-fed lambs without access to pasture; G4H, indoor-fed lambs with daily access to pasture for four hours; G8H, indoor-fed lambs with daily access to pasture for eight hours.

**Figure 5 animals-13-02644-f005:**
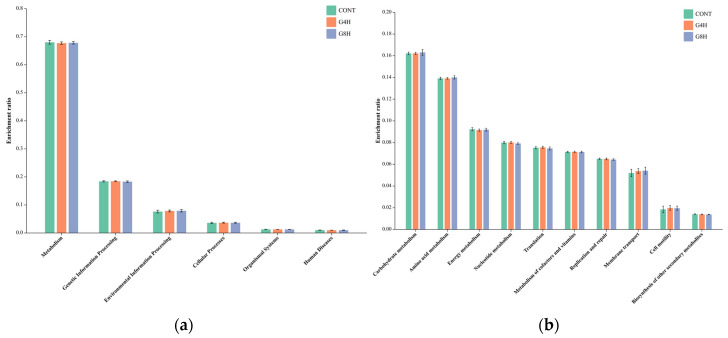
Comparison of the KEGG functional prediction by primary (**a**) and secondary (**b**) pathway between Huang-huai sheep with different feeding regimes. CONT, indoor-fed lambs without access to pasture; G4H, indoor-fed lambs with daily access to pasture for four hours; G8H, indoor-fed lambs with daily access to pasture for eight hours.

**Figure 6 animals-13-02644-f006:**
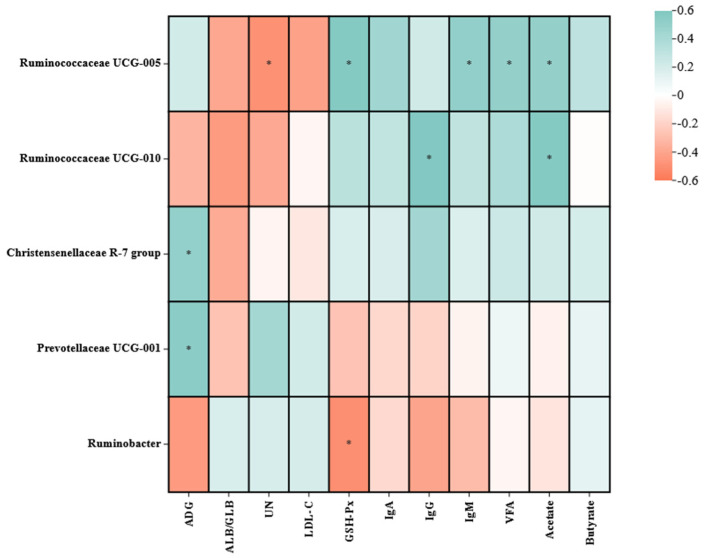
Heat map summarizing Pearson’s correlations between different rectal microbiomes and average daily gain (0–90 d), serum metabolites, and rumen VFA. The color intensity and circle size are proportional to the correlation values; * 0.01 < *p* ≤ 0.05. ADG, average daily gain; ALB/GLB, albumin/globulin; Ig A, Immunoglobulin A; Ig G, Immunoglobulin G; Ig M, Immunoglobulin M; LDL-C, low-density lipoprotein cholesterol; UN, urea nitrogen; VFA, volatile fatty acids.

**Table 1 animals-13-02644-t001:** Component of dietary formula and chemical composition (% dry matter).

Ingredients	Content (% Dry Matter)	Chemical Composition	Content (% Dry Matter)
Whole corn silage	45.00	Dry matter	46.8
Corn	30.20	Crude protein	16.35
Soybean meal	14.00	Neutral detergent fiber	30.28
Wheat bran	7.80	Acid detergent fiber	18.29
NaHCO_3_	1.00	Crude ash	8.28
Salt	1.00	Calcium	0.71
Premix ^(1)^	1.00	Phosphorus	0.40
Total	100		

Note ^(1)^: Each 1 kg premix contained 3500 IU vitamin A, 1000 IU vitamin D, 40 IU vitamin E, 50 mg iron, 40 mg manganese, 10 mg copper, 40 mg Zinc, and 0.3 mg selenium.

**Table 2 animals-13-02644-t002:** Effects of different feeding regimes on growth performance of Huang-huai lambs.

Title ^1^	Treatment			Mean	SEM	*p*-Value
	CONT ^2^	G4H	G8H			
Initial BW ^1^, kg 0 d	29.71	30.79	30.36	30.29	1.22	0.945
Final BW, kg						
30 d	36.43	39.86	37.43	37.91	1.25	0.555
60 d	39.43	44.59	41.57	41.86	1.51	0.416
90 d	42.36	49.14	45.34	45.61	1.57	0.235
ADG, g/day						
0–30 d	223.81	302.38	235.71	253.97	14.89	0.069
30–60 d	100.00	157.62	138.10	131.91	15.29	0.328
60–90 d	97.62	151.90	125.71	125.08	21.44	0.625
0–90 d	140.48	203.97	166.51	170.32	11.51	0.080

^1^ ADG, average daily gain; BW, body weight; SEM, standard error of the mean. ^2^ CONT, indoor-fed lambs without access to pasture; G4H, indoor-fed lambs with daily access to pasture for four hours; G8H, indoor-fed lambs with daily access to pasture for eight hours.

**Table 3 animals-13-02644-t003:** Effects of different feeding regimes on serum biochemical indices in Huang-huai lambs.

Title ^1^	Treatment			Mean	SEM	*p*-Value	Normal Reference Range ^3^
	CONT ^2^	G4H	G8H				
TP (g/L)	60.74	61.96	60.96	61.22	1.57	0.951	59.0–78.0
ALB (g/L)	23.86	22.79	22.41	23.02	0.57	0.592	27.0–37.0
GLB (g/L)	36.88	39.18	38.55	38.20	1.13	0.721	35.0–57.0
ALB/GLB	0.65	0.59	0.59	0.61	0.01	0.087	_
UN (mmol/L)	6.36	5.56	5.37	5.76	0.19	0.096	4.6–15.7
ALT (IU/L)	19.15	16.79	14.05	16.66	1.36	0.337	19.6–44.1
AST (IU/L)	113.21	101.64	99.48	104.78	4.05	0.362	49.0–123
TC (mmol/L)	1.68	1.84	1.90	1.81	0.09	0.623	1.4–5.2
TG (mmol/L)	0.30	0.24	0.26	0.27	0.02	0.640	_
HDL-C (mmol/L)	0.75	0.82	0.84	0.80	0.02	0.187	_
LDL-C (mmol/L)	0.76 ^a^	0.62 ^b^	0.63 ^b^	0.67	0.03	0.042	_

^a,b^ means within a row with different subscripts differ when *p* < 0.05. ^1^ ALB, albumin; ALT, alanine aminotransferase; AST, aspartate aminotransferase; GLB, globulin; HDL-C, high-density lipoprotein cholesterol; LDL-C, low-density lipoprotein cholesterol; SEM, standard error of the mean; TC, total cholesterol; TG, triglyceride; TP, total protein; UN, urea nitrogen. ^2^ CONT, indoor-fed lambs without access to pasture; G4H, indoor-fed lambs with daily access to pasture for four hours; G8H, indoor-fed lambs with daily access to pasture for eight hours. ^3^ Source: normal reference range from [37,38,39].

**Table 4 animals-13-02644-t004:** Effects of different feeding regimes on serum antioxidant and immunity indices in Huang-huai lambs.

Item ^1^	Treatment			Mean	SEM	*p*-Value
	CONT ^2^	G4H	G8H			
Antioxidant indice						
GSH-Px (pg/mL)	1071.43 ^b^	1384.35 ^a^	1544.85 ^a^	1333.54	51.82	<0.001
Immunoglobulin indices						
IgA (ug/mL)	118.34 ^b^	157.63 ^a^	181.71 ^a^	152.56	7.44	<0.001
IgG (mg/mL)	22.98 ^b^	29.84 ^a^	35.42 ^a^	29.41	1.73	0.007
IgM (ug/mL)	1176.23 ^b^	1565.52 ^a^	1681.90 ^a^	1474.55	59.12	<0.001

^a,b^ means within a row with different subscripts differ when *p* < 0.05. ^1^ GSH-Px, glutathione peroxidase; IgA, Immunoglobulin A; IgG, Immunoglobulin G; IgM, Immunoglobulin M. ^2^ CONT, indoor-fed lambs without access to pasture; G4H, indoor-fed lambs with daily access to pasture for four hours; G8H, indoor-fed lambs with daily access to pasture for eight hours.

**Table 5 animals-13-02644-t005:** Effects of different feeding regimes on ruminal pH, NH_3_-N and total VFAs content (mmol L^−1^), molar proportions of individual VFA (mmol L^−1^) in Huang-huai lambs.

Item	Treatment			Mean	SEM	*p*-Value
	CONT ^1^	4H	8H			
pH	6.39	6.28	6.65	6.44	0.094	0.326
NH_3_-N (mg/dL)	13.08	13.12	15.75	13.98	0.787	0.322
Total VFA (mmol L^−1^)	44.39	53.34	48.66	48.80	1.676	0.100
Acetate (mmol L^−1^)	27.75	33.05	31.80	30.87	0.948	0.057
Propionate (mmol L^−1^)	7.65	9.75	8.32	8.57	0.524	0.280
Butyrate (mmol L^−1^)	6.95	8.57	6.36	7.29	0.404	0.071
Valerate (mmol L^−1^)	0.88	0.84	0.99	0.90	0.108	0.862
Isobutyrate (mmol L^−1^)	0.88	0.93	0.86	0.89	0.045	0.825
Isovalerate (mmol L^−1^)	0.28	0.20	0.33	0.27	0.068	0.766
Acetate: propionate	3.72	3.47	4.23	3.81	0.215	0.378

^1^ CONT, indoor-fed lambs without access to pasture; G4H, indoor-fed lambs with daily access to pasture for four hours; G8H, indoor-fed lambs with daily access to pasture for eight hours.

**Table 6 animals-13-02644-t006:** Effects of different feeding systems on the alpha diversity indices related to the fecal microbiota of Huang-huai lambs.

Item	Treatment			Mean	SEM	*p*-Value
	CONT ^1^	G4H	G8H			
Chao 1	1354.79	1538.33	1551.71	1481.61	39.81	0.144
PD_whole_tree	78.74	86.72	87.36	84.27	2.35	0.459
Goods_coverage	1.00	0.99	1.00	1.00	0.0002	0.245
Observed_features	1224.17	1360.83	1399.17	1328.06	34.53	0.135
Shannon index	7.90	8.28	8.43	8.20	0.098	0.182
Simpson index	0.99	0.99	0.99	0.99	0.001	0.103

^1^ CONT, indoor-fed lambs without access to pasture; G4H, indoor-fed lambs with daily access to pasture for four hours; G8H, indoor-fed lambs with daily access to pasture for eight hours.

## Data Availability

The data are available from the corresponding author upon request.

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
