# Peer review of "Stall-Feeding of Sheep on Restricted Grazing: Effects on Performance and Serum Metabolites, Ruminal Fermentation, and Fecal Microbiota"

_animals, 2023, doi:10.3390/ani13162644_

Round 1
Reviewer 1 Report
Dear Editor,
the text contains many inaccuracies and/or careless errors. I have highlighted many areas where English expression could be improved.
Information about the statistical model should be added. Equations used would be appreciated.
In summary, paper will need of major revision.
Line 22: please, delete “i.e.”
Line 22 check CONT
Line 25-26: Rephrase the average daily gain specifying that is not significative.
Line 34-35 Re-evaluate the phrase “. Overall, these results suggest that indoor feeding with restricted grazing time can improve fattening performance,”
Line 52: replace comma by punctuation.
Line 53 specify FBW at the first use.
Line 72 delete moreover (repeated word)
Line 97 check the phrase; it seems an unconscious pun
Line 104 replace “were fed” with “grazed” if it is true.
Line 104 delete one comma.
Line 108 use the correct format to indicate the hour.
Line 107 “CONT”or “CON”?
Line 120 replace level with composition and specify the unit of measure (%DM, g/1000 g…..i.e.)
Line 142 delete one comma.
Line 137 delete parenthesis.
Line 197 “Significance was determined at a level of P ≤ 0.05, 197 and 0.05 < P ≤ 0.10 was considered as a tendency towards a difference”. Please delete the second part of the phrase referred to a tendency towards a difference. It could be reported a tendency of the value in the discussion.
Specify the statistical models used: for example, fixed factor…..
Equation will be appreciated.
Line 301, 302, 305, 308 and follow CONT or CON. This error is repeated along all the text.
Line 303 delete one “was”.
Line 356 In my opinion those results were not similar because no statistical differences emerged from different rearing system, while only a tendency could be noted. I think that those should be underline. Add information about other research: sex of animal, breed, age……
In the same sense also the phrase at line 369-372 it’s not fully true in your study.
L436 what is referred “s”?
Dear Authors,
text contains many inaccuracies and/or careless errors. I have highlighted many areas where English expression could be improved.
Reviewer 2 Report
The manuscript compared grazing and confined animals. The title needs to be adjusted, it became a methodology title, and without making it clear what the main focus of the research is. Rephrase.
I would like to begin by congratulating you on the number of variables analyzed. However, many variables really do not have a clear explanation of why they were evaluated. Remember that to choose variables, you need to have a hypothesis of what might happen.
it was not clear to me what the pasture was during the period and how this management was carried out; how the height of this pasture was monitored; Is there a pasture consumption hypothesis?
Microbiota is a very interesting analysis, and the results here are difficult to be observed due to the poor quality of the figure. Need to make it visible.
Author Response
请参阅附件。

Round 2
Reviewer 1 Report
Dear Authors,
The document has been improved. Nevertheless, there are yet present some not well-presented paragraph. Information regard statistical analysis are incomplete, or it seem that some factors were not considered in experimental design.
Also, discussion could be more punctual to obtained results.
In summary, this paper will yet need a major revision. However, specific comment are reported below.

Minor revisions are necessary
Author Response
Point 1: Abstract: avoid the use of “show a tendency”. Please respect the limit of significance pointed to P ≤ 0.05. Please, view comment of line 223. If the significance is pointed on a level of P ≤ 0.05, it is necessary respect this limit. Response 1: As suggested by the reviewer, we changed from “Average daily gain showed a tendency (P =0.081) to be higher for G4H group compared with CONT group.” to “Average daily gain were not affected (P =0.081) by different feeding systems and averaged 170.32 g/day.” at the Lines 25-26 of the revised manuscript. Point 2: Line 78 use past simple verb Response 2: As suggested by the reviewer, we changed from “regimes have a direct” to “regimes had a direct.” at the Line 78 of the revised manuscript. Point 3: Line 97 specify the duration of the experiment “April to July 2021 (4 months) Response 3: As suggested by the reviewer, we changed “April to July 2021” to “3 April to 14 July 2021” at line 97 of the revised manuscript. Point 4: Line 103 reformulate “Twenty-four 4 to 5-month-old female Huang-huai” Response 4: Thanks for your insightful comment. As suggested by the reviewer, we changed from “Twenty-four 4 to 5-month-old female Huang-huai sheep with an initial body weight of 28.96 ± 1.04 kg (mean ± SD) were randomly chosen ” to “Twenty-four female Huang-huai sheep, approximately 5 months old with an initial body weight of 28.96 ± 1.04 kg (mean ± SD), were randomly chosen” at the Lines 103-104 of the revised manuscript. Point 5: How many animals for group? How were distributed the animals for age class? Response 5: In this study, each treatment group had eight lambs at line 105 of the revised manuscript. Thanks for your very relevant comment. Experimental lambs were mainly born between late November and early December 2020. It is a great pity that we have not recorded the exact date of birth of every lamb. The sheep were randomly divided into three groups. Point 6: Line 117-122 add am or pm in the specific time slot. Response 6: As suggested by the reviewer, we changed “(08:30 and 18:30 h), (18:30 h), 09:00 to 13:00 h, 14:00 to 18:00 h and 08:30 to 12:30 h and from 14:00 to 18:00 h” to “(8:30 a.m. and 6:30 p.m.), (6:30 p.m.), 9:00 a.m. to 1:00 p.m., 2:00 p.m. to 6:00 p.m., 8:30 a.m. to 12:30 p.m. and from 2:00 p.m. to 6:00 p.m..” at lines 117-123 of the revised manuscript, respectively. Point 7: Line 129 Please, in table 1 title or in the table report the unit of measure respectively for ingredients and chemical composition. Replace nutritional level with chemical composition. Response 7: As suggested by the reviewer, we changed “nutrient levels and nutritional level” to “chemical composition ” at line 128 and Table 1 of the revised manuscript, respectively. In addition, we added “(% dry matter)” in table 1 of the revised manuscript. Point 8: Line 135 specify the average age±standard deviation when lambs. Response 8: Thanks for your insightful comment. It is with great regret that we have not recorded the exact date of birth of every lamb. This comment is valuable and very helpful for our future research. Point 8: Materials and methods: -Improve the statistical analysis and deepen basing on comment at line 286, Response 5: As suggested by the reviewer, we added “Venn analysis was performed on the number of OTUs obtained.” at lines 191-192 of the revised manuscript, respectively. Point 9: -Use the specific format for equation (Author instruction) Response 9: Thanks for your insightful comment. We perused the manuscript that had been published in Animals magazine. As suggested by the reviewer, We corrected the computational model, and changed “Y i = μ + D i + E i, where Yi is the dependent variable, μ is mean vale, D i means the fixed effect of feeding system, and E i means standard error.”to “Y i = μ + D i + E, where Y i is the dependent variable, μ is mean vale, D i means the fixed effect of feeding system, and E means the random effect”at lines 210-212 of the revised manuscrip, according to the method of Zhao et al. (2023) and Kinh et al. (2023). Zhao, Y.; Zhang, Y.; Bai, C.; Ao, C.; Qi, S.; Cao, Q.; Erdene, K. Effects of the dietary inclusion of allium mongolicum regel extract on serum index and meat quality in Small-Tailed Han sheep. Animals 2023, 13, 110. Kinh, L.V.; Vasanthakumari, B.L.; Sugumar, C.; Thanh, H.L.T.; Thanh, N.V.; Wealleans, A.L.; Ngoan, L.D.; Loan, N.V.T.H. Effect of a combination of lysolecithin, synthetic emulsifier and monoglycerides on the apparent ileal digestibility, metabolizable energy and growth performance of growing pigs. Animals 2023, 13, 88. Point 10: - Line 193 title of 2.6 delete “calculation and” Response 10: As suggested by the reviewer, we deleted “calculation and” at lines 206 of the revised manuscript, respectively. Point 11: - Line 210-213 Y ij = μ + D i + e ij Why did you not consider animal effect? I suggest you consider animal effect as nested into feeding system. Response 11: Thanks for your insightful comment. We perused the manuscript that had been published in Animals magazine. As suggested by the reviewer, We corrected the computational model, and changed “Y i = μ + D i + E i, where Yi is the dependent variable, μ is mean vale, D i means the fixed effect of feeding system, and E i means standard error.”to “Y i = μ + D i + E, where Y i is the dependent variable, μ is mean vale, D i means the fixed effect of feeding system, and E means the random effect”at lines 210-212 of the revised manuscrip, according to the method of Zhao et al. (2023) and Kinh et al. (2023). Zhao, Y.; Zhang, Y.; Bai, C.; Ao, C.; Qi, S.; Cao, Q.; Erdene, K. Effects of the dietary inclusion of allium mongolicum regel extract on serum index and meat quality in Small-Tailed Han sheep. Animals 2023, 13, 110. Kinh, L.V.; Vasanthakumari, B.L.; Sugumar, C.; Thanh, H.L.T.; Thanh, N.V.; Wealleans, A.L.; Ngoan, L.D.; Loan, N.V.T.H. Effect of a combination of lysolecithin, synthetic emulsifier and monoglycerides on the apparent ileal digestibility, metabolizable energy and growth performance of growing pigs. Animals 2023, 13, 88. Point 12: Why did you not consider age in the model? Age should be considered as covariate because animals had different age from 4 to 5 months when the experiment started. Response 12: Thanks for your insightful comment. In this study, all lambs were born between late November and early December 2020. It is with great regret that we have not recorded the exact date of birth of every lamb. This comment is valuable and very helpful for our future research. Point 13: Line 220 “During the three-months of the experiment” from April to July were more than three months. Response 13: As suggested by the reviewer, we changed “April to July 2021” to “3 April to 14 July 2021” at line 97 of the revised manuscript. Point 13: If the significance is pointed on a level of P ≤ 0.05, it is necessary respect this limit. Line 223 please don’t use “a tendency of significance” here and in the rest of article. You can replace for example by “the ADG tend to…… even if the significance were not achieved (P=0.08)” In my opinion there were not statistical references that pointed the use of 0.05 < P ≤ 0.10 as a tendency towards a difference. If references is available it could be pointed. Response 13: Thanks for your very relevant comment. However, there is precedent in other published papers for Values between p > 0.05 and p < 0.10 were considered a significant trend (Ospina-Romero, et al., 2023; Thanh et al., 2023 and Hu et al., 2023). Ospina-Romero, M.A.; Medrano-Vázquez, L.S.; Pinelli-Saavedra, A.; Sánchez-Villalba, E.; Valenzuela-Melendres, M.; Martínez-Téllez, M.Á.; Barrera-Silva, M.Á.; González-Ríos, H. Productive performance, physiological variables, and carcass quality of finishing pigs supplemented with ferulic acid and grape pomace under heat stress conditions. Animals 2023, 13, 2396. Thanh, L.P.; Loor, J.J.; Mai, D.T.T.; Hang, T.T.T. Effect of fish oil and linseed oil on intake, milk yield and milk fatty acid profile in goats. Animals 2023, 13, 2174. Hu, Y.; Tang, S.; Zhao, W.; Wang, S.; Sun, C.; Chen, B.; Zhu, Y. Effects of dried blueberry pomace and pineapple pomace on growth performance and meat quality of broiler chickens. Animals 2023, 13, 2198. Point 14: Line 226- 131 The value 131.9 g/ d are not present in the table or I’m not seeing…. Response 14: Thank you for your good suggestion. As suggested by the reviewer, we added mean value in all tables at the revised manuscript. Point 15: Line 286 in statistical analysis did not emerge that you performed also a Venn analysis. Response 15: As suggested by the reviewer, we added “As shown in Venn diagram (Figure 1),” at line 282 of the revised manuscript, respectively. Point 16: Figure 2. This is the figure of a principal component analysis! From figure it seems that you consider only 6 animals for each group, while I understood that animals were 24, 8 for each group. Response 16: Thanks for your very relevant comment. In this study, we randomly selected six lambs from each experimental group, according to the method of Yin et al. (2021) and Pang et al. (2022). Yin, X.; Ji, S.; Duan, C.; Ju, S.; Zhang, Y.; Yan, H.; Liu, Y. Rumen fluid transplantation affects growth performance of weaned lambs by altering gastrointestinal microbiota, immune function and feed digestibility. Animal 2021, 15, 100076. Pang, K.; Yang, Y.; Chai, S.; Li, Y.; Wang, X.; Sun, L.; Cui, Z.; Wang, S.; Liu, S. Dynamics changes of the fecal bacterial community fed diets with different concentrate-to-forage ratios in Qinghai yaks. Animals 2022, 12, 2334. Point 17: Figure 6 in the text there were not referring to figure 6 Response 17: As suggested by the reviewer, we added “Figure 6” at line 356 of the revised manuscript, respectively. Point 18: Discussion: -in my opinion discussion are too long. Some part could be considered a superfluous introduction to the discussion paragraphs. Response 18: As suggested by the reviewer, we deleted “However, increased grazing time as adopted for animals in G8H lambs resulted in higher energy digestion due to increased physical activity.”, “UN content is a key parameter related to nitrogen metabolism, and reflects in vivo protein metabolism status. [45].”, “However, LDL-C was easily oxidized by foam cells, which leads to cardiovascular diseases [46]”, and “The activity of both transaminases significantly increased when the liver was damaged or when protein metabolism was enhanced” at lines 399-407 of the revised manuscript, respectively. Point 19: Line 369 and line 371 had. Response 19: As suggested by the reviewer, we changed “have” to “had” at the Line 370 and line 372 of the revised manuscript, respectively. Point 20: Line 372 “another study” insert reference. Response 20: Thanks for your insightful comment. We added “[42]” at line 376 of the revised manuscript, respectively. Point 21: Line 304 and all discussion text: please continue to use past simple verb (had, contributes…..) Response 21: Thanks for your good comments. As suggested by the reviewer, we changed past simple verb at the discussion sections of the revised manuscript. Point 22: Line 384 this phrase is unlinked to results of the study, check or rephrase. Response 22: As suggested by the reviewer, we deleted “However, increased grazing time as adopted for animals in G8H lambs resulted in higher energy digestion due to increased physical activity.” at the revised manuscript. Point 23: Line 425 suggested Response 23: As suggested by the reviewer, we changed “suggests” to “suggested” at line 421 of the revised manuscript, respectively. Point 24: Line 452 delete have Response 24: As suggested by the reviewer, we deleted “have” at the revised manuscript. Point 25: Line 466 delete showed a tendency and replace by tended. Response 25: As suggested by the reviewer, we changed “ showed a tendency ” to “tended” at line 462 of the revised manuscript, respectively. Point 26: Line 512 suggested vs suggest. However, I suggest deleting this phrase because it is approximately. Response 26: As suggested by the reviewer, we deleted “ Taken together, these results further suggest that Huang-huai sheep were suitable for various feeding systems. ” at the revised manuscript, respectively.
Reviewer 2 Report
Adjustments were made and the doubtful points clarified. Publication favorable.
Author Response
Response to Reviewer 2 Comments
Special thanks for your approval of the revision of our manuscript.
Additional minor revisions were made to the manuscript using track changes to improve English and readability.
Yours sincerely,
Yafeng Huang
No. 130, West Changjiang Road, Hefei, Anhui, China, 230036
